# The Role and Mechanism of Retinol and Its Transformation Product, Retinoic Acid, in Modulating Oxidative Stress-Induced Damage to the Duck Intestinal Epithelial Barrier In Vitro

**DOI:** 10.3390/ani13193098

**Published:** 2023-10-04

**Authors:** Li Zhang, Rui Tang, Yan Wu, Zhenhua Liang, Jingbo Liu, Jinsong Pi, Hao Zhang

**Affiliations:** 1Institute of Animal Husbandry and Veterinary Science, Hubei Academy of Agricultural Sciences, Wuhan 430064, China; zl9588623@163.com (L.Z.); ruiq1207515167@163.com (R.T.); wuyanwh@163.com (Y.W.); liangzhenghua2046@163.com (Z.L.); pijinsong@sina.com (J.P.); 2School of Life Science and Engineering, Southwest University of Science and Technology, Mianyang 621010, China; liuswust@163.com; 3Hubei Key Laboratory of Animal Embryo Engineering and Molecular Breeding, Wuhan 430064, China

**Keywords:** retinol, retinoic acid, duck intestinal epithelial cell, oxidative stress, mechanism

## Abstract

**Simple Summary:**

In the intensive farming process of laying ducks, the intestinal tract is susceptible to oxidative stress, which results in the impairment of intestinal barrier function. This study investigated the effects of retinol and its transformation product, retinoic acid, on duck intestinal cells under oxidative stress. Cells were treated with different ratios of retinol and retinoic acid, and their viability, antioxidant capacity, and barrier function were assessed. The expression of genes related to oxidative stress and barrier function was measured. The results showed that treatment with different ratios of retinol and retinoic acid improved cell viability, antioxidant capacity, and barrier function. High retinoic acid treatment enhanced cell activity and barrier function by affecting certain pathways, while high retinol treatment promoted barrier repair through different pathways. This study highlights the roles of retinol and retinoic acid in protecting duck intestinal epithelial cells from oxidative stress and enhancing barrier function and provides insights into their underlying mechanisms. This understanding could contribute to developing strategies to mitigate oxidative damage and improve intestinal health.

**Abstract:**

This study aimed to investigate the effects and mechanisms of retinol and retinoic acid on primary duck intestinal epithelial cells under oxidative stress induced by H_2_O_2_. Different ratios of retinol and retinoic acid were used for treatment. The study evaluated the cell morphology, viability, antioxidative capacity, and barrier function of cells. The expression of genes related to oxidative stress and the intestinal barrier was analyzed. The main findings demonstrated that the treated duck intestinal epithelial cells exhibited increased viability, increased antioxidative capacity, and improved intestinal barrier function compared to the control group. High retinoic acid treatment improved viability and gene expression, while high retinol increased antioxidative indicators and promoted intestinal barrier repair. Transcriptome analysis revealed the effects of treatments on cytokine interactions, retinol metabolism, PPAR signaling, and cell adhesion. In conclusion, this study highlights the potential of retinol and retinoic acid in protecting and improving intestinal cell health under oxidative stress, providing valuable insights for future research.

## 1. Introduction

China is a major producer and consumer of duck meat and eggs [1]. Intensive breeding, which is considered more efficient and cost-effective, can help reduce the environmental pollution caused by traditional duck farming [2]. However, intensive systems of duck farming, such as those involving heat stress, immune challenges, and high-density feeding, can cause stress responses that lead to the production of a large amount of reactive oxygen species and free radicals. Heat stress, dysbiosis, leaky gut syndrome, and mycotoxins are the main “secret killers” in poultry that lead to chronic oxidative stress and inflammation, which in turn impact health and animal performance. Additionally, chronic stress in poultry is linked to the emergence of antimicrobial resistance (AMR), which the WHO has recently identified to be among the most important problems threatening human health globally, increasing the demand for safe antimicrobials to treat the collateral damage resulting from dysbiosis [3]. All forms of chronic stress, regardless of origin, negatively impact the chicken’s overall performance, health, and welfare [4]. These substances can cause lipid peroxidation and damage to membranes, proteins, and other large biological molecules in the intestinal mucosa, including DNA [5,6,7]. To repair intestinal mucosal damage, animals activate their endogenous antioxidant system in conjunction with exogenous antioxidants that protect their body against oxidative stress induced free radicals and other reactive oxygen species. This has been extensively studied in humans and animal models [8,9,10,11]. However, the activation of the antioxidant system exhibits species-specific and tissue-specific characteristics, and the underlying interaction between oxidative stress and intestinal function in ducks remains uncertain.

Vitamin A is a fat-soluble vitamin and an essential nutrient for animals [12]. The term “vitamin A” encompasses a group of chemically related organic compounds that includes retinol, retinal, and retinoic acid (RA) [13]. The concentration of vitamin A in the blood can be disturbed under pathological or physiological conditions [14,15]. In addition to dietary levels that can change vitamin A concentration, key tissues in the animal body, such as the liver, use retinol dehydrogenase and retinaldehyde dehydrogenase to irreversibly convert retinol into retinoic acid through enzymatic reactions [16]. Changes in the concentration of vitamin A have been demonstrated to have various biological effects in the intestine, mainly enhancing immune function, altering the proliferation and maturation of intestinal cells, and promoting the expression of tight junction proteins [17,18]. Zheng et al. reported a significant alteration in the endogenous concentration of plasma retinoic acid during the stress-sensitive period caused by changes in the rearing environment of ducks [19]. Therefore, we believe that changes in the concentration of vitamin A play an important role in maintaining the health of the duck intestinal barrier.

The role of different forms of vitamin A is controversial. Some studies believe that RA is the major active metabolite of vitamin A that regulates a range of biological processes [20] and that retinol is biologically inactive and must be converted into its bioactive form, all-trans retinoic acid [21]. However, other studies have shown that retinol is functional [22]. Its activity, antioxidative effect, and role and mechanism in intestinal barrier function in duck intestinal epithelial cells (dIECs) are less studied. The present study aimed to investigate the effect of different ratios of retinol and RA on dIEC status, oxidative stress indices, and a range of intestinal barrier-related indicators and to conduct transcriptome sequencing to explore the molecular mechanism of their effects in duck intestinal epithelial cells challenged with H_2_O_2_.

## 2. Materials and Methods

### 2.1. Animals and Ethics Statement

The procedures were conducted according to the Guidelines for Experimental Animals Challenged established by the Ministry of Science and Technology. All procedures involving animal subjects were approved by the Animal Ethics Committee of the Hubei Academy of Agricultural Sciences (2021-620-000-001-021).

### 2.2. dIEC Culture

The dIECs were primary cells isolated from six Nonghu No. 2 laying duck embryos at embryonic day 26, which were incubated together in the incubator. The abdominal cavity was opened under sterile conditions, and the duodenum and jejunum were isolated and placed in a 90 mm Petri dish with 8 mL of warm PBS (37 °C) (Gibco, Thermo Fisher Scientific, Grand Island, NY, USA). After moving mesentery and blood vessels, these tissue fragments were digested with PBS containing 20 mg of type I collagenase (Gibco, Thermo Fisher Scientific, Grand Island, NY, USA) for 70 min at 37 °C while shaking. After discarding the digestive juices, the tissue fragments were rinsed with PBS multiple times, and the cell suspension was collected and centrifuged. The cells were resuspended in complete medium, filtered using a 100 μm nylon membrane, and seeded into 12-well cell culture dishes (Sangon, Shanghai, China) at a density of 3 × 10^5^ cells/cm^2^. Cells were grown in Dulbecco’s modified Eagle’s medium/F12 (Sigma-Aldrich, St. Louis, MO, USA) containing 5% FBS (Mediatech. Inc., Manassas, VA, USA), 1% penicillin-streptomycin (Sigma-Aldrich, St. Louis, MO, USA), and 100 ng/mL epidermal growth factor (Rocky Hill, NJ, USA). Cells were cultured in a 5% CO_2_ and 37 °C incubator for 24 h.

### 2.3. Cell Treatment and Morphology

Primary cultured IECs that were placed in a 6-well microplate (2 × 10^6^ cells/well) (Sangon, Shanghai, China) were stimulated with 50 μmol/L H_2_O_2_ for 4 h to establish an oxidative model. The dIECs under oxidative stress were divided into the following five groups: the negative control, the positive control, and 3 treated groups. According to the total amount of vitamin A in the serum, the treated groups with different ratios of retinol and retinoic acid, included treated group 1 (TG1) (retinol/retinoic acid = 0.1 μM:0.3 μM), treated group 2 (TG2) (retinol/retinoic acid = 0.2 μM:0.2 μM), and treated group 3 (TG3) (retinol/retinoic acid = 0.3 μM:0.1 μM). Both retinol and RA were dissolved in DMSO. The negative group (CG1) was not treated with any substance but had DMSO added to the culture medium, and the positive group (CG2) had oxidative stressed dIECs treated with DMSO. Each treatment had 3 replicates. The condition of the cells was observed using an inverted microscope (AX R with NSPARC. Nikon, Shanghai, China) and photographed at 200× after 24 h of treatment. Normal cell morphology appears cobblestone-like, with minimal intercellular space, a low number of dead cells, and a rare occurrence of vacuoles. Conversely, any deviation from this is considered abnormal cell morphology.

### 2.4. Cell Viability

The treated cells were digested with enzymes from 6-well plates and seeded in 96-well plates (2 × 10^3^ cells/well). After the completion of seeding, 20 μL of Cell Counting Kit-8 (CCK-8; BS350A, Biosharp, Hefei, China) was added to each well. After incubation for 4 h, the absorbance was measured at 450 nm with a microplate reader (Victor X5, PerkinElmer, Singapore). The cell viability percentage was calculated using the following formula: cell viability = (mean absorbance in the test wells)/(mean absorbance in the blank control wells) × 100%. Cell viability was determined after 12 and 24 h of treatment.

### 2.5. Biochemical Determinations

After completion of the treatments, the medium was removed, and the cells were washed twice with PBS. The cells were collected in a 1.5 mL centrifuge tube, followed by the addition of 200 μL of RIPA lysis buffer. After incubating on ice for 30 min, the mixture was centrifuged at 12,000 rpm for 10 min; afterward, the supernatants were collected. The content of malondialdehyde (MDA), activity of total antioxidant capacity (T-AOC), and total superoxide dismutase (T-SOD) were measured by the colorimetric method (UV-2550, Shimadzu, Kyoto, Japan) with commercial assay kits (Nanjing Jiancheng Institute of Bioengineering, Nanjing, China), which were used in accordance with the manufacturer’s instructions. Biochemical measurements were performed 24 h after treatment.

### 2.6. Monolayer Integrity

The transepithelial electrical resistance (TEER) was measured using an epithelial volt-ohm meter with a chopstick electrode (Millicell ERS-2, Millipore, Billerica, MA, USA). The cells were seeded in Transwell^®^ 6-well plates (5 × 10^4^ cells/well) (Costar 3412, Corning, Grand Island, NY, USA). The electrode was immersed at a 90° angle with one tip in the basolateral chamber and the other in the apical chamber. The TEER value was calculated as follows: TEER value (Ω∙cm^2^) = (cell growth hole valueb−lank hole value) × micropore area. The microwell area of the insert area of the Transwell^®^ 6-well cell culture plate was 4.67 cm^2^. The TEER was measured at both 12 h and 24 h after treatment.

### 2.7. RNA Extraction

Total RNA was isolated from the cells at the end of treatment using TRIzol^®^ (Invitrogen, Carlsbad, CA, USA) and then treated with RQ1 DNase (Promega, Madison, WI, USA). RNA integrity was assessed using the Agilent Bioanalyzer 2100 system (Agilent Technologies, Santa Clara, CA, USA). cDNA was synthesized using a Thermo^®^ RtAid First Strand cDNA Synthesis Kit (Thermo Fisher Scientific, Waltham, MA, USA).

### 2.8. Real-Time Quantitative Polymerase Chain Reaction (RT–qPCR)

Quantitative real-time PCR was performed using a LightCycler^®^ 96 system (Roche, Life Science) with the SYBR^®^ PrimeScriptTM RT–qPCR Kit (Roche, Mannheim, Germany). In Table 1, we present the primer sequences designed using premier 5 for genes associated with the composition of tight junction proteins, the natural antioxidant cytoprotective system, and RNA-seq identified differentially expressed genes (DEGs). These primers are intended for subsequent experimental applications, including the analysis of the cellular state and the validation of RNA-seq results. Relative mRNA expression was calculated using the 2^−ΔΔCt^ method and normalized to β-actin.

### 2.9. RNA Sequencing

Three samples were selected randomly from each replication per treatment for sequencing. An NEB Next Ultra RNA Library Preparation Kit (Illumina, San Diego, CA, USA) was used to prepare a sequence library with 3 μg RNA for each sample. Poly-T oligomer magnetic beads were employed to purify mRNA, which was then fragmented and reverse-transcribed into cDNA. Single-end sequencing was performed on the HiSeq 2000 platform by Novogene Inc. (Tianjin, China).

### 2.10. Identification of Differentially Expressed Genes (DEGs)

The identification of differentially expressed genes (DEGs) was carried out using HISAT2 software 2.2.1.0 to align the clean reads to the reference genome (ZJU1.0). Gene expression levels were quantified using the fragments per kilo bases per million mapped reads (FPKM) method. The screening of DEGs was performed using the DESeq command of DESeq2 Software 4.2, with the following criteria for filtering DEGs: (1) |log2(FoldChange)| > 1; (2) padj ≤ 0.05; and (3) genes with an FPKM value greater than or equal to 1 in both groups.

### 2.11. Functional Enrichment Analysis

GO term enrichment analysis and Kyoto Encyclopedia of Genes and Genomes (KEGG) (http://www.genome.jp/kegg, accessed on 20 November 2021) pathway enrichment analysis were conducted for genes with upregulated and downregulated expression using the R package cluster profiler [23]. For the GO term, completely annotated genes of ducks were appointed as the background. KEGG was used to identify the metabolic pathways or signal transduction pathways that were significantly enriched with DEGs by comparing them to the entire genome background. Only the GO terms or KEGG pathways with *p*-value < 0.05 were considered significantly enriched [24]. All RNA-seq data were deposited into the Gene Expression Omnibus (GEO) database from NCBI (https://www.ncbi.nlm.nih.gov/geo, accessed on 20 November 2021).

### 2.12. Statistical Analysis

Data analysis was performed using SPSS 23.0 software (IBM, Inc., Armonk, NY, USA), with variance analysis conducted using one-way ANOVA. Multiple comparisons of the mean values between groups were carried out using Duncan’s method, and the results are presented as the mean ± SD. Statistical significance was considered at *p* < 0.05.

## 3. Results

### 3.1. Survival of Duck Intestinal Epithelial Cell after Oxidative Damage

Cell phenotype observations showed that the cells in CG1 exhibited a healthy state (Figure 1A). There were fewer floating dead cells, and there were very few vacuoles between the cells. However, the cells in CG2 displayed increased vacuolation, and the number of floating dead cells increased due to oxidative stress-induced apoptosis or decreased vitality (Figure 1B). In contrast, the treatment groups (Figure 1C–E) showed reduced vacuolation between cells compared to the CG2 group. In particular, the high-concentration retinoic acid group (TG1) exhibited fewer floating dead cells, and the cells maintained a normal morphology (Figure 1C). In the TG2 and TG3 groups, floating cell fragments were observed, indicating visible signs of repair. However, it should be noted that some cell gaps were also present in these groups (Figure 1D,E).

The cell viability analysis conducted after 12 h of treatment revealed a significant decrease in cell viability in the positive group (CG2) (*p* < 0.05). Compared to the positive group (CG2), treatment with varying proportions of retinol and retinoic acid significantly increased oxidative stress cell viability at 12 h. However, despite the increase in cell viability in the positive group at 24 h, only TG1 exhibited higher cell viability (Figure 1F).

### 3.2. Indices of Redox Status

As shown in Table 2, significant differences in T-AOC, T-SOD, and MDA levels were observed among the groups (*p* < 0.05). Compared to the CG1 group, the levels of T-AOC and T-SOD in the TG1 group showed a significant decrease, while the level of MDA significantly increased (*p* < 0.05). The CG2 group showed significantly lower T-AOC levels than the CG1, TG1, TG2, and TG3 groups. No significant differences were observed among the groups given different ratios of retinol and retinoic acid.

The positive group showed significantly lower T-SOD levels than all other groups. The T-SOD activity of TG1 was lower than that of the other treatment groups.

The negative control (CG1) group showed significantly lower MDA levels than the other groups. The CG2 group showed significantly higher MDA levels than groups TG1 and TG3. No significant differences were observed among the groups given different ratios of retinol and retinoic acid.

### 3.3. TEER of Intestinal Epithelial Cell Monolayers

After 12 and 24 h of treatment, the addition of different ratios of retinol and RA (TG1, TG2, and TG3) significantly increased TEER compared to that in the positive group (*p* < 0.05) (Figure 2). Specifically, the high-concentration retinol group (TG3) exhibited significantly higher TEER values than the groups treated with retinol and retinoic acid at equal ratios (TG2) after 12 and 24 h of treatment. Additionally, at 12 h of treatment, the TG3 group showed a significant increase in TEER compared to the high-concentration retinoic acid group (TG1), while this difference was not significant at 24 h.

### 3.4. Transcripts of Antioxidative System-Related Genes and Tight Junction-Related Genes after Oxidative Challenge

As shown in Figure 3, the expression of antioxidative system-related genes and tight junction-related genes in the CG2 group was significantly reduced compared to that in the CG1 group. As illustrated in Figure 3A,B, cells treated with different ratios of retinol and RA exhibited increased expression levels of Nrf2 and HO-1, which are genes associated with the cellular antioxidant response, compared to the positive group cells (*p* < 0.05). Moreover, an increase in the ratio of RA led to a further upregulation of the relative expression of Nrf2 and HO-1 (*p* < 0.05). Figure 3C–E demonstrates that the expression of the tight junction-related genes ZO-1, claudin-1, and Occludin was significantly upregulated (*p* < 0.05) in cells treated with different ratios of retinol and RA compared to that in the positive group. Furthermore, the gene expression levels of ZO-1 and claudin-1 in TG1 were higher than those in the other treatment groups. There were no significant differences in the expression levels of the Occludin gene among groups given different ratios of retinol and retinoic acid.

### 3.5. Differential Expression of mRNAs in Duck Intestinal Epithelial Cells

The volcano plots in Figure 4A–F show the number of up- and downregulated genes of interest, including those in different treatment groups compared to the positive group (Figure 4A–C), as well as in groups treated with retinol and retinoic acid at various ratios (Figure 4D,E). In addition, the plots also display other combinations of treatment groups and controls that we analyzed (Figure 4F).

The bar graphs in Figure 4G illustrate the number of DEGs in the different groups analyzed. Our analysis revealed that increasing the ratio of retinoic acid led to a higher number of DEGs. Specifically, treatment with retinol and retinoic acid at a 1:3 ratio resulted in a greater number of DEGs than the other treatments. These results suggest that retinoic acid plays a crucial role in regulating gene expression and that its effects may be dose dependent, with a nonlinear relationship between the amount of retinoic acid and the number of DEGs.

Venn diagram analysis of DEGs among the treatment groups revealed that 391 DEGs were found between TG1 and TG2, while 108 genes were DEGs between TG2 and TG3. Among these genes, 10 genes were found to be expressed in both comparison groups, whereas 381 genes were specifically expressed in TG1 vs. TG2, and 98 genes were specifically expressed in TG2 vs. TG3 (Figure 4H; Appendix A).

Furthermore, the Venn diagram analysis of DEGs between each treatment group and the positive control demonstrated that among the 1917 differentially expressed genes in CG2 vs. TG1, 629 DEGs were shared among all three groups while 891 DEGs were unique. Additionally, 181 DEGs and 239 DEGs were exclusively found in CG2 vs. TG2 and CG2 vs. TG3, respectively (Figure 4I; Appendix A).

### 3.6. Validation of Differentially Expressed Messenger RNAs

DEGs associated with cellular functions were selected for further validation using RT–qPCR analysis to confirm the findings from RNA-seq. Validation confirmed the expression of the claudin-1 gene, which is involved in intestinal barrier function, the HO-1 gene, which is involved in the cellular antioxidant response, and genes (PPARG, ACSL1, and ACSL5) related to the PPAR signaling pathway. To facilitate the comparison of gene expression levels obtained through different methods, the FPKM (fragments per kilobase of transcript per million mapped reads) values from RNA-seq were normalized using the FPKM value of the CG1 group as a reference value of 1. The expression levels of these genes in other groups were represented as the ratio of their FPKM values to the FPKM value of the CG1 group. The expression trends of PPARG and ACSL5 were found to be consistent between high-throughput sequencing and qRT–PCR (Figure 5A,B). Therefore, the results obtained from RNA-seq can be utilized for subsequent functional validation of the identified genes.

### 3.7. Gene Ontology Annotation and Kyoto Encyclopedia of Genes and Genomes Enrichment Analysis of Differentially Expressed Genes

#### 3.7.1. Comparisons between the Untreated and Treated Groups

Gene Ontology (GO) includes biological processes (BP), cellular components (CC), and molecular functions (MF). GO terms with a *p* < 0.05 were considered significantly enriched. Figure 6A lists the top 30 GO terms of CG vs. TG (the samples that were not treated with retinol and retinoic acid and the samples that were treated with retinol and retinoic acid), including BP, CC, and MF. The BP functions related to cell adhesion and the G-protein-coupled receptor signaling pathway were significantly enriched. The extracellular matrix was the enriched GO term for CC. In the MF category, the DEGs were enriched in GO terms including G-protein-coupled receptor activity, transmembrane signaling receptor activity, signaling receptor binding and molecular transducer activity. All the DEGs of CG vs. TG were mapped to the KEGG database to further investigate DEGs involved in key pathways (Figure 6B), such as cell adhesion molecules (CAMs), cytokine–cytokine receptor interactions, neuroactive ligand–receptor interactions, retinol metabolism, glycine, serine and threonine metabolism, and the PPAR signaling pathway.

#### 3.7.2. Comparisons between the Positive Control and Different Treatment Groups

Figure 7A lists the top 30 GO terms of DEGs of CG2 vs. TG1. The BP functions related to microtubule-based movement, cell adhesion and the movement of cell or subcellular components were significantly enriched. Extracellular regions, the extracellular matrix and extracellular region parts were enriched GO terms for CC. In the MF category, the DEGs were significantly enriched in GO terms including oxidoreductase activity, acting on paired donors, incorporation or reduction in molecular oxygen, signaling receptor binding, tetrapyrrole binding, heme binding, iron ion binding, and growth factor activity. A total of 1917 DEGs of CG2 vs. TG1 were significantly categorized into 5 pathways that mapped to the KEGG database. The DEGs were determined to be involved in key pathways (Figure 7B), including CAMs, the cell cycle, ferroptosis, ECM-receptor interaction, cytokine–cytokine receptor interaction, vascular smooth muscle contraction, and glycine, serine and threonine metabolism.

Figure 7C lists the top 30 enriched GO terms of CG2 vs. TG2 DEGs. The BP functions were related to the regulation of cell growth. The extracellular matrix, extracellular region, extracellular region parts, supramolecular complex, supramolecular polymer, supramolecular fiber, intermediate filament, intermediate filament cytoskeleton, and polymeric cytoskeletal fiber were enriched GO terms for CC. In the MF category, the DEGs were significantly enriched in GO terms including metallopeptidase activity, metalloendopeptidase activity, tetrapyrrole binding, heme binding, insulin-like growth factor binding, growth factor binding and G-protein-coupled receptor activity. A total of 1189 DEGs of CG2 vs. TG2 were significantly categorized into 7 pathways that mapped to the KEGG database. The DEGs were determined to be involved in key pathways (Figure 7D), including retinol metabolism, the PPAR signaling pathway, CAMs, ECM-receptor interaction, and neuroactive ligand–receptor interaction.

Figure 7E lists the top 30 GO terms enriched in DEGs of CG2 vs. TG3. The CCs were related to the extracellular region, extracellular matrix and extracellular region parts. In the MF category, the DEGs were significantly enriched in GO terms including signaling receptor binding, receptor regulator activity, receptor ligand activity, metalloendopeptidase activity, and cytokine receptor binding. A total of 1162 DEGs of CG2 vs. TG3 were significantly categorized into 5 pathways that mapped to the KEGG database. The DEGs were determined to be involved in key pathways (Figure 7F), including cytokine–cytokine receptor interaction, CAMs, retinol metabolism, the PPAR signaling pathway and ECM-receptor interaction.

#### 3.7.3. Comparisons between Different Treatment Groups

Figure 8A lists the top 30 significant GO terms enriched in DEGs of TG1 vs. TG2. The CCs related to the extracellular region were significantly enriched. In the MF category, the DEGs were enriched in GO terms including calcium ion binding, heme binding, tetrapyrrole binding, chemokine activity, and chemokine receptor binding. A total of 391 DEGs of TG1 vs. TG2 were categorized into 3 pathways that mapped to the KEGG database. The DEGs were determined to be involved in key pathways (Figure 8B), including the Toll-like receptor signaling pathway, the neuroactive ligand–receptor interaction and melanogenesis.

Figure 8C lists the top 30 significant GO terms enriched in DEGs of TG2 vs. TG3. The BP functions related to the immune response, immune system process, and defense response were significantly enriched. The extracellular region was the enriched GO term for CC. In the MF category, the DEGs were enriched in GO terms including cytokine activity, chemokine activity, chemokine receptor binding, receptor regulator activity and receptor ligand activity. A total of 108 DEGs of TG2 vs. TG3 were categorized into 10 pathways that mapped to the KEGG database. The DEGs were determined to be involved in key pathways (Figure 8D), such as cytokine–cytokine receptor interaction, the RIG-I-like receptor signaling pathway, the NOD-like receptor signaling pathway, salmonella infection, the Toll-like receptor signaling pathway, the cytosolic DNA-sensing pathway, the AGE-RAGE signaling pathway in diabetic complications, pantothenate and CoA biosynthesis, and arginine biosynthesis.

## 4. Discussion

Intensive farming practices induce oxidative stress, resulting in damage to the mucosal lining of the duck intestine. Hydrogen peroxide-induced injury to animal intestinal epithelial cells serves as a well-established model for studying intestinal epithelial damage and repair [25,26]. Our previous investigation demonstrated reduced cell viability and decreased superoxide dismutase (SOD) activity in dIECs after H_2_O_2_ treatment, thus confirming the suitability of this model for studying oxidative stress [27]. Therefore, we employed the hydrogen peroxide-induced intestinal barrier injury model to investigate the mechanisms underlying barrier repair. Retinol, also known as vitamin A, plays a crucial role as a promoter of antioxidants within the body. It functions as a precursor to various biologically active compounds, such as retinal and retinoic acid, which are essential for maintaining the health and integrity of our cells and tissues. Retinol aids in the production of antioxidant enzymes such as superoxide dismutase and catalase, which help neutralize harmful free radicals and reactive oxygen species (ROS) [28]. The concentration of vitamin A varies between 0.5 and 20 μM in animal serum [29]. Different concentrations of vitamin A have different effects on the repair of intestinal barrier injury [20]. Our preliminary findings indicated that higher concentrations of RA and retinol were toxic to dIECs. Hence, we carefully selected appropriate working concentrations for this study to elucidate the effects of the biological conversion of retinol to retinoic acid on oxidatively-stressed dIECs.

Evaluating the cell state and barrier function of dIECs involves the measurement of a variety of key indicators, such as cell viability, oxidative stress markers, transepithelial electrical resistance (TEER), and the expression levels of key functional genes. Our results of cell viability, oxidative stress markers, transepithelial electrical resistance, and the expression of key genes related to tissue antioxidant and intestinal cell tight junction function showed that the different ratios of retinol/RA treatment all showed good effects on improving cell activity and enhancing intestinal barrier function compared to the positive group. This is similar to the research results of Yamada, Xiao, etc. [30,31], which provides us with a repair strategy to address duck intestinal barrier damage in production.

However, when further analyzing the effects of different retinol and RA ratio treatments on cells, we found that the effects of high-concentration RA treatment and high-concentration retinol treatment on the cell state were different. The cell phenotype provides the most intuitive assessment of the cell state [32]. Studies have found that during physiological and pathological injury processes, retinol and its metabolites can better maintain the state of intestinal epithelial cells [33]. Notably, the high-concentration retinoic acid group (TG1) exhibited a healthier cellular phenotype with fewer floating dead cells and the formation of rule-governed ‘paving stone’ shaped cells, while the high-concentration retinol group (TG3) cells showed tighter aggregation. The CCK-8 assay is a quantitative indicator for evaluating cell viability [34,35]. Our experimental results show that the high proportion RA group had more effective improvement of the viability of dIECs, and this was more effective in the early stage of treatment. Lukonin et al.’s research shows that the retinoic acid signaling pathway can regulate intestinal cell fate transitions and help maintain the balance between regeneration and homeostasis [36]. This implies that the addition of retinoic acid at a higher concentration had a more pronounced effect on promoting cell viability in dIECs. In terms of activating dIEC activity, the role of retinol was relatively minor.

Interestingly, when detecting cell antioxidant indicators, we found that within the concentration range of our experiment, the T-AOC level and T-SOD level of the high-concentration retinol group (TG3) cells were significantly higher than those of the TG1 cells, as they showed better antioxidant capacity. In addition, as a direct indicator of barrier function, the TEER barrier function test results also showed that the TEER value of the high-concentration retinol group (TG3) was higher, especially after 12 h of treatment. These results suggest that when the ratio of retinol is higher, dIECs have stronger antioxidant capacity, and retinol also improves the barrier function of dIECs. However, the qPCR results showed that compared with the other treatment groups, the expression of key genes related to the intestinal barrier and antioxidant capacity in the TG1 group was higher. Gene expression results do not exhibit a perfect correlation with TEER measurements and antioxidant capacity, suggesting that transcriptional regulation mediated by RA through RXR or other nuclear receptors may lead to the upregulated expression of certain genes in TG1. While there is a significant association between transcriptional regulation and actual tight junctions/antioxidative processes, it is not entirely congruent. In the high-retinol group (TG3), despite the relatively low qPCR abundances of specific genes, these genes may still influence intercellular tight junctions and antioxidant capabilities through alternative mechanisms. For example, retinol can form a complex with the retinol binding protein (RBP) [37], and this complex stimulates recognition through the retinoic acid 6 (STRA6) receptor, which mediates the absorption of extracellular retinol into the cytoplasm [38]. The related mechanism may affect the biological function of retinol absorbed into the tissue [39].

To further elucidate the phenotypic results, we analyzed the gene changes in dIECs in response to the treatments through high-throughput sequencing. This analysis aids in clarifying the reasons for the changes in dIEC status caused by different transformations of retinol. By comparing the DEGs between TGs and GCs, we found that the addition of retinol triggered changes in the expression of key genes in the extracellular matrix and extracellular region of dIECs. Numerous studies have shown that retinol and related retinoic acid signals can alter the expression of genes related to the extracellular matrix, significantly affecting intercellular interactions [40,41] and promoting the maintenance of barrier function [42]. KEGG pathway enrichment results showed that DEGs were mainly enriched in the CAM pathway, further confirming these findings. KEGG pathway enrichment analysis also revealed that DEGs were enriched in pathways such as G-protein-coupled receptor activity, cytokine–cytokine receptor interaction, glycine-serine-threonine metabolism, and retinol metabolism. Among these, G-protein-coupled receptor activity plays a crucial role in maintaining cell proliferation and migration [43], especially in the intestine, where leucine-rich repeat-containing G-protein-coupled receptor 5+ (LGR5+) stem cells can promote intestinal epithelial renewal. The cytokine–cytokine receptor pathway can participate in the maintenance of barrier function [44,45]. Aon et al. found that glycine-serine-threonine metabolism can also cooperate with other signaling pathways to enhance cell activity [46]. The activation of the retinol metabolism pathway in dIECs further proves that vitamin A can work as a biological antioxidant, serving as a barrier of defense for the intestinal mucosa against free radicals [47]. These results confirm that the gene expression changes and pathway activations caused by the addition of different transformations of vitamin A are important ways for vitamin A to enhance dIEC activity, strengthen antioxidant capacity, and maintain intestinal barrier function.

We analyzed the DEGs of different retinol/RA ratio groups and the positive control group. The results further confirmed that RA is an important transcriptional regulator. The higher the concentration of retinoic acid, the more DEGs, and high concentrations of RA enhanced transcriptional regulation ability, which is closely related to transcriptional regulation with nuclear receptors (RXR, RAR) [48]. However, the number of DEGs in TG2 vs. CG2 and TG3 vs. CG2 was similar, indicating that the enhancement of RA’s transcriptional regulation ability is not linear, but when RA reaches a certain level, it can significantly increase the number of DEGs. This result can also be reflected by the number of DEGs in TG3 vs. TG2 and TG2 vs. TG1. Chen’s research results show that the level of transcription regulation by different concentrations of RA is nonlinear [49]. KEGG pathway enrichment analysis results showed that high-concentration retinoic acid treatment (TG1) specifically activated the cell cycle, ferroptosis, glycine, serine and threonine metabolism, and other signaling pathways. In addition to the glycine, serine, and threonine metabolism mentioned above and the cell activity-related pathways, promoting the cell cycle [50] and ferroptosis resistance [51] are also important ways to promote cell proliferation and improve cell vitality.

The KEGG pathways enriched by the DEGs between TG3 and CG2 and the DEGs between TG1 and CG2 and TG2 and CG2 showed no significant difference. However, by comparing the DEGs of TG1 vs. TG2 and TG2 vs. TG3, we found that the number of DEGs in TG1 vs. TG2 was large but mainly enriched in the ligand–receptor binding KEGG pathway. Although the number of DEGs in TG2 vs. TG3 was only 108, these DEGs were significantly enriched in 10 KEGG pathways. Among them, the RIG-I-like receptor (retinoic acid-inducible gene-I-like receptors, RLRs) signaling pathway, the NOD-like receptor (nucleotide-binding oligomerization domain-like receptors, NLRs) signaling pathway, Salmonella infection, the cytosolic DNA-sensing pathway, the AGE-RAGE signaling pathway in diabetic complications, pantothenate and CoA biosynthesis, arginine biosynthesis, and influenza A were all specifically activated. Research has shown that the AGE-RAGE signaling pathway can modulate gut permeability [52]. Arginine biosynthesis plays a role in promoting wound healing [53]. The NOD-like receptor signaling pathway is related to intestinal barrier function and epithelial barrier function [54,55]. The activation of these pathways may be the reason why the dIECs in the TG3 group showed better barrier function. Pantothenate and CoA biosynthesis have antioxidant effects [56,57], so the activation of this pathway may be an important way for high retinol to improve cell antioxidant capacity. The RIG-I-like receptor signaling pathway, Salmonella infection [58], influenza A [59], and cytosolic DNA-sensing pathway are signaling pathways related to innate immunity and antibacterial, anti-inflammatory, and antiviral activities. The activation of these pathways indicates that the addition of high retinol may also promote innate immunity and improve the immune ability of the intestinal mucosa to resist bacteria and viruses.

## 5. Conclusions

The metabolic changes in vitamin A in animals are complex processes. This study determined the effects of different proportions of retinol and RA on the phenotype and gene expression of dIECs within the concentration range we studied and analyzed the signaling pathways enriched by DEGs between groups. The metabolic changes in this study related to vitamin A increased antioxidative indicators and promoted intestinal barrier repair. The research results can provide ideas for the nutritional regulation of duck intestinal barrier oxidative stress damage repair and provide direction and materials for the study of the endogenous regulatory mechanism of vitamin A.

## Figures and Tables

**Figure 1 animals-13-03098-f001:**
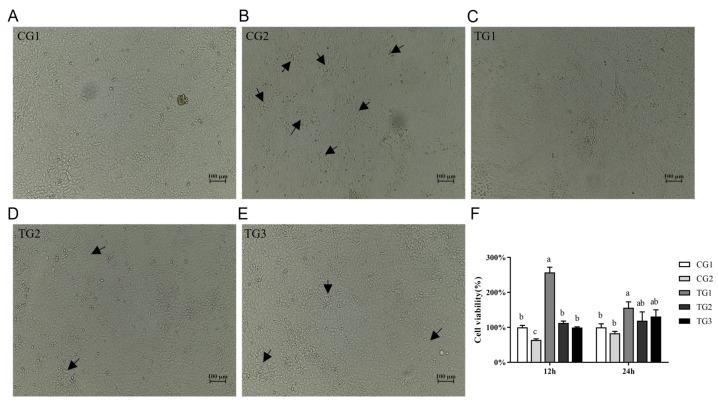
Cell viability analysis. (**A**–**E**) Morphological changes in dIECs after challenge with H_2_O_2_ and subsequent different ratios of retinol and retinoic acid treatment (×100). The arrows in the figure indicate cell floatation and intercellular space. (**F**) Comparison of cell viability at 12 h and 24 h after H_2_O_2_ treatment. The data are presented as the mean ± standard deviation (*n* = 3). Bars with different letters indicate significant difference (*p* < 0.05).

**Figure 2 animals-13-03098-f002:**
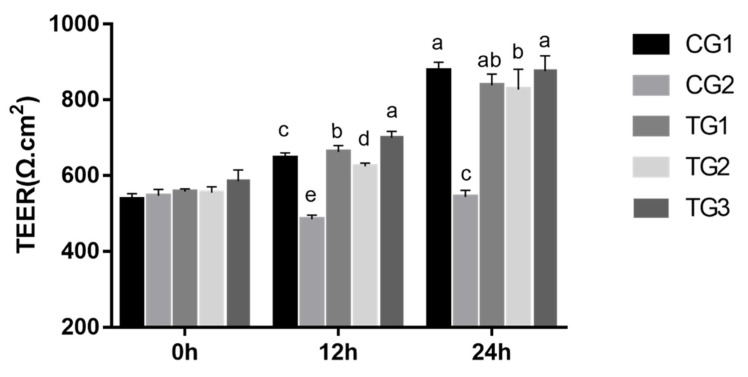
Effects of different ratios of retinol and retinoic acid treatment on the TEER of dIECs. The data are presented as the mean ± standard deviation (*n* = 3). Bars with different letters indicate significant difference (*p* < 0.05).

**Figure 3 animals-13-03098-f003:**
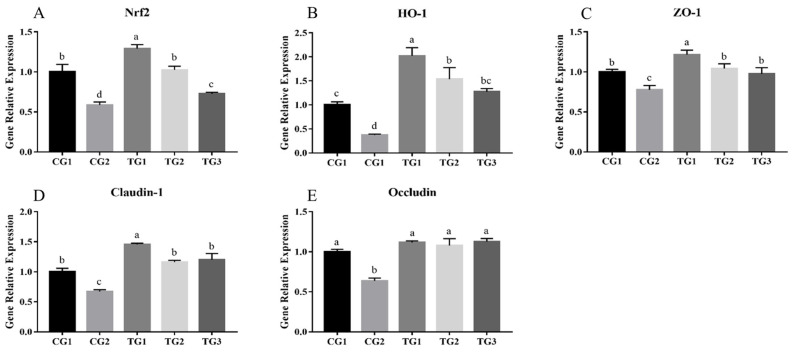
Effect of different ratios of retinol and retinoic acid treatment on transcripts of anti-oxidative stress, endoplasmic reticulum stress and intestinal barrier-related genes after oxidative stress induced by H_2_O_2_. (**A**) The relative gene expression levels of Nrf2 in different treatment groups. (**B**) The relative gene expression levels of HO-1 in different treatment groups. (**C**) The relative gene expression levels of ZO-1 in different treatment groups. (**D**) The relative gene expression levels of Claudin-1 in different treatment groups. (**E**) The relative gene expression levels of Occludin in different treatment groups. The data are presented as the mean ± standard deviation (*n* = 3). Bars with different letters indicate significant difference (*p* < 0.05); Nrf2: nuclear factor (erythroid 2)-like 2; HO-1: heme oxygenase-1; ZO-1: zonula occludens-1.

**Figure 4 animals-13-03098-f004:**
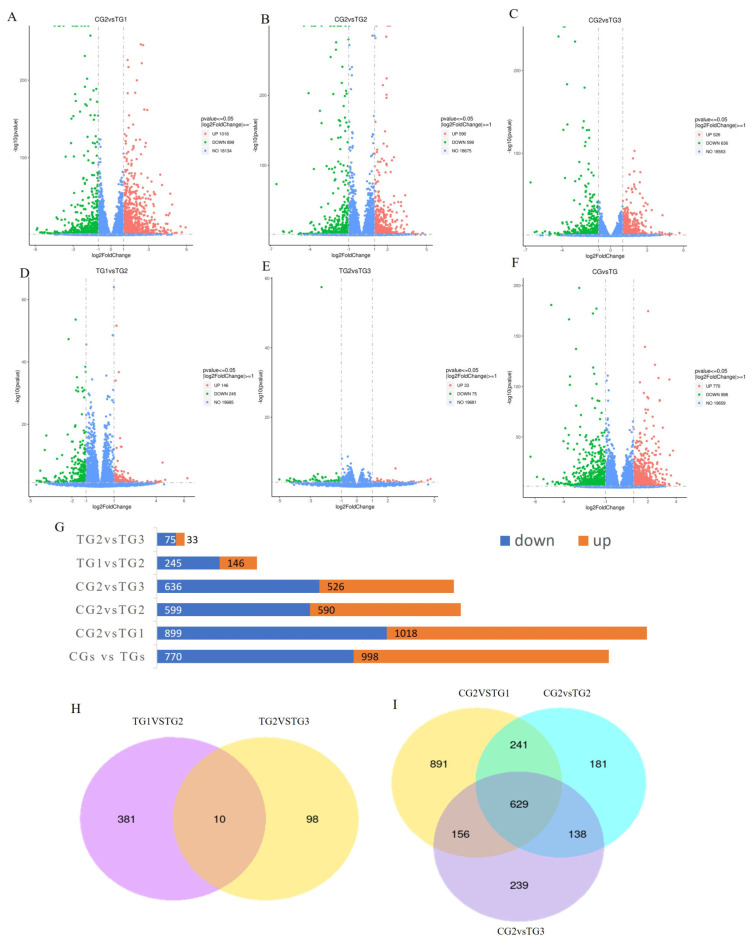
Analysis of differentially expressed genes. (**A**–**C**) Volcano plot of differentially expressed genes in different treatment groups compared to both the positive groups. (**D**,**E**) Comparison of retinol and retinoic acid treatment groups at various ratios. (**F**) Comparison of retinol and retinoic acid treatment and control groups. The *x*-axis and *y*-axis indicate log2 (fold change) and 2log10 (FDR) of differentially expressed genes, respectively. The red color represents upregulated genes, and the green color represents downregulated genes. (**G**) Comparison bar chart of differentially expressed genes in different groups. The red color represents upregulated genes, and the blue color represents downregulated genes. (**H**,**I**) Venn diagram analysis of differentially expressed genes in different treatment groups.

**Figure 5 animals-13-03098-f005:**
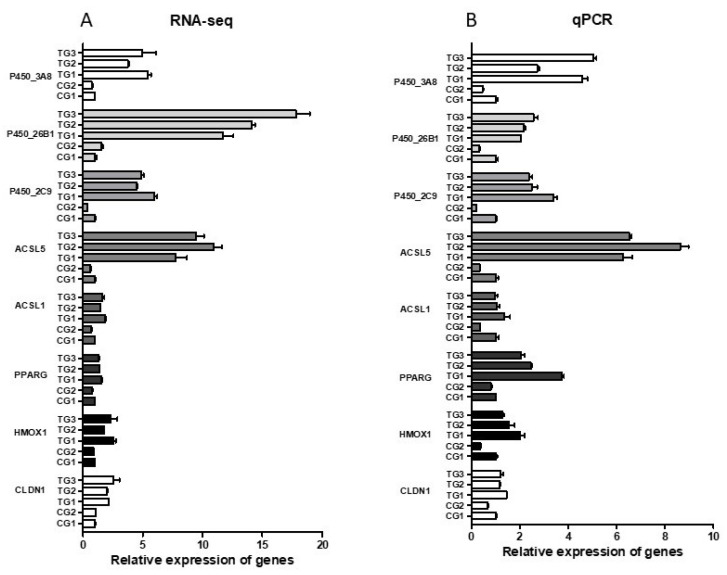
RT-qPCR validated the expression patterns of candidate genes involved in cellular functions. (**A**) RNA-seq of selected gene expression levels related to cellular function. (**B**) The relative expression was detected by RT-qPCR. The data are presented as the mean ± standard deviation (*n* = 3). Abbreviations: RNA-seq, RNA sequencing; RT-qPCR, quantitative real-time PCR.

**Figure 6 animals-13-03098-f006:**
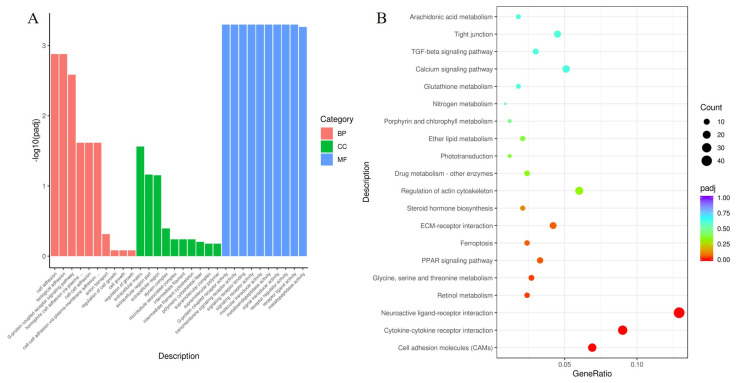
GO and KEGG enrichment analysis of differentially expressed genes between the untreated groups and treated groups. (**A**) Top 30 enriched GO terms of differentially expressed genes in the CG group compared with the TG group. (**B**) Top 20 enriched KEGG terms of differentially expressed genes in the CG group compared with the TG group. Abbreviations: GO, Gene Ontology; KEGG, Kyoto Encyclopedia of Genes and Genomes.

**Figure 7 animals-13-03098-f007:**
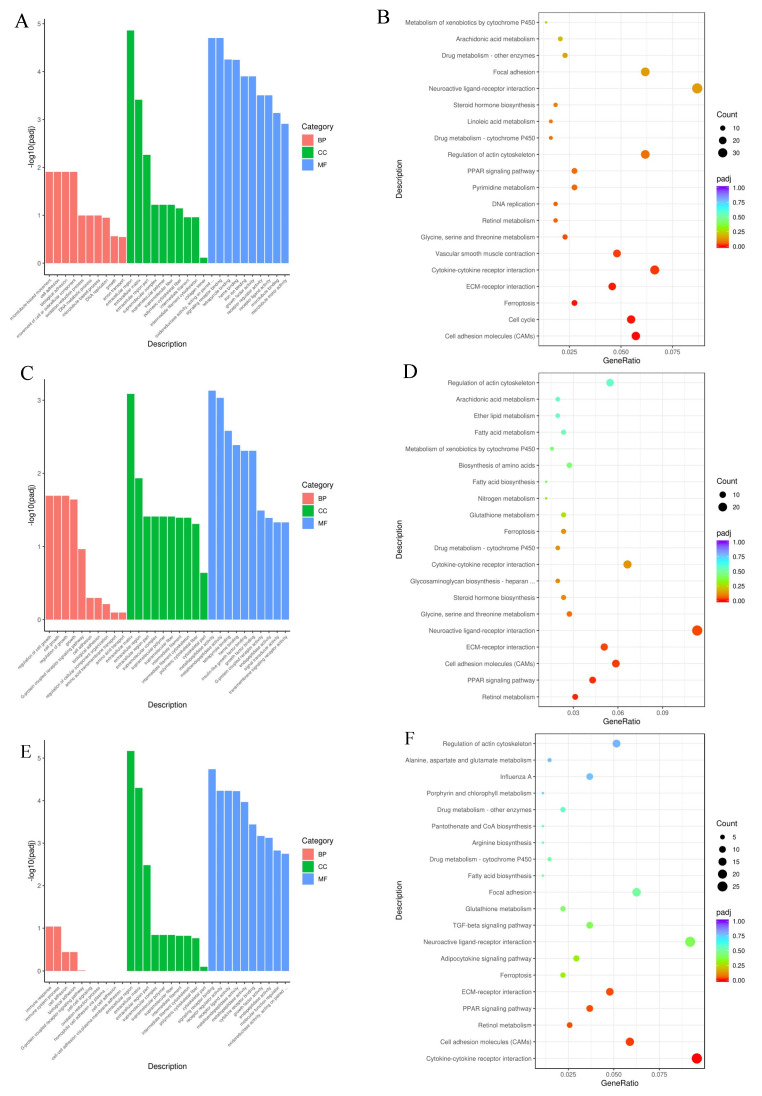
GO and KEGG enrichment analysis of differentially expressed genes between the positive control and different treatment groups. (**A**,**B**) Top 30 enriched GO terms and top 20 enriched KEGG terms of differentially expressed genes in the CG2 group compared with the TG1 group. (**C**,**D**) Top 30 enriched GO terms and top 20 enriched KEGG terms of differentially expressed genes in the CG2 group compared with the TG2 group. (**E**,**F**) Top 30 enriched GO terms and top 20 enriched KEGG terms of differentially expressed genes in the CG2 group compared with the TG3 group. Abbreviations: GO, Gene Ontology; KEGG, Kyoto Encyclopedia of Genes and Genomes.

**Figure 8 animals-13-03098-f008:**
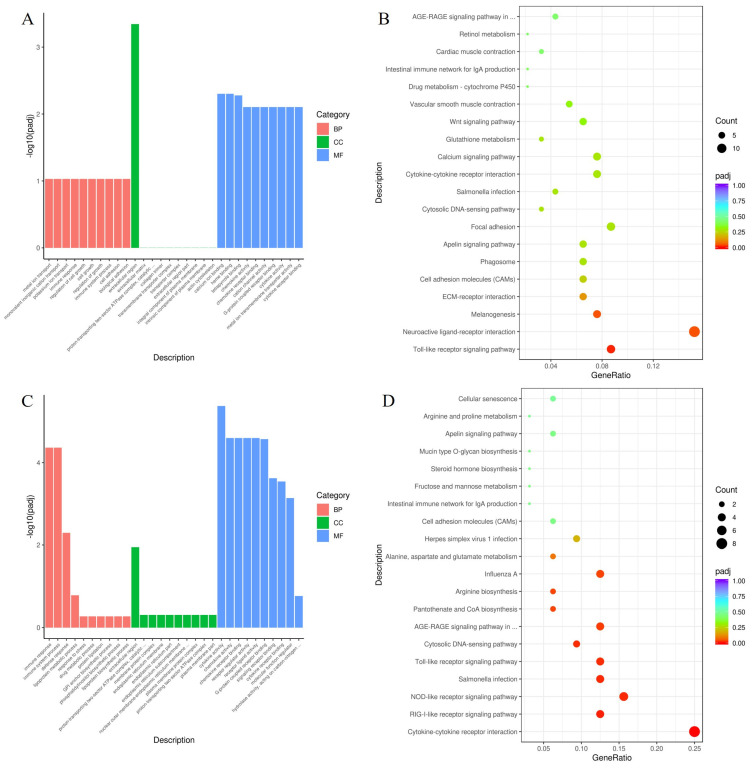
GO and KEGG enrichment analysis of differentially expressed genes between the different treatment groups. (**A**,**B**) Top 30 enriched GO terms and top 20 enriched KEGG terms of differentially expressed genes in the TG1 group compared with the TG2 group. (**C**,**D**) Top 30 enriched GO terms and top 20 enriched KEGG terms of differentially expressed genes in the TG2 group compared with the TG3 group. Abbreviations: GO, Gene Ontology; KEGG, Kyoto Encyclopedia of Genes and Genomes.

**Table 1 animals-13-03098-t001:** The primers used in this study.

Gene Name	Primer Sequence (5′ to 3′)
Nrf2	Sense: TGTTGAATCATCTGCCTGTG Antisense: TTGTGAACGGTGCTTTGG
HO-1	Sense: TGCCTACACTCGCTATCTGG Antisense: CGTTCTCCTGGCTCTTTGA
ZO-1	Sense: GCACCGAAGCCTACACTCA Antisense: CGGTAATACTCTTCATCTTCTT
Claudin-1	Sense: TGACCAGGTGAAGAAGATGC Antisense: GGGTGGGTGGATAGGAAGT
Occludin	Sense: GCTGGGCTACAACTACGGGT Antisense: ACGATGGAGGCGATGAGC
PPARG	Sense: TAACGCTCCTGAAATACGGT Antisense: GAACTTCACAGCGAACTCAA
ACSL1	Sense: CTCTGCGTTACTCCACCG Antisense: GCATAGCATCCCTGTTCG
ACSL5	Sense: AACCCAACCAACCTTATC Antisense: TGTCACAAATCACTACGC
P4502C9	Sense: CCAGGTGAAACCAAAGGA Antisense: GAGCAAACCGACGGACAT
P4503A8	Sense: GCCAAGTTCAATGTAAGCG Antisense: CCAGTTCGTAAGCCAGGTAA
P45026B1	Sense: ACGGGAGAAGTACGGGAACG Antisense: TGGATGTCGCCGATGGAG
β-actin	Sense: GCTATGTCGCCCTGGATTT Antisense: GGATGCCACAGGACTCCATAC

**Table 2 animals-13-03098-t002:** Effects of retinol/retinoic acid on antioxidant capacity.

	CG1	CG2	TG1	TG2	TG3
T-AOC (mM/mL)	25.98 ± 2.46 ^a^	17.89 ± 0.75 ^c^	20 ± 1.07 ^b^	24.80 ± 3.82 ^ab^	24.13 ± 1.92 ^ab^
T-SOD (U/mL)	465.42 ± 38.93 ^ab^	339.54 ± 13.88 ^d^	397.15 ± 24.92 ^c^	456.15 ± 6.45 ^ab^	466.71 ± 27.16 ^a^
MDA (nmol/mL)	3.39 ± 0.06 ^c^	4.90 ± 0.07 ^a^	4.35 ± 0.45 ^b^	4.64 ± 0.31 ^ab^	4.42 ± 0.12 ^b^

Results presented as mean ± standard deviation (*n* = 3); the table shows that values with same superscript letters in the same line are of no significant difference (*p* > 0.05), those lacking a common superscript differ (*p* < 0.05); T-AOC: total antioxidant; T-SOD: total superoxide dismutase; MDA: malondialdehyde.

## Data Availability

The datasets of RNA-Seq generated for this study can be found in the Sequence Read Archive (SRA) database (Bioproject ID: PRJNA757931).

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
