# Peer review of "The Role and Mechanism of Retinol and Its Transformation Product, Retinoic Acid, in Modulating Oxidative Stress-Induced Damage to the Duck Intestinal Epithelial Barrier In Vitro"

_animals, 2023, doi:10.3390/ani13193098_

Round 1
Reviewer 1 Report
Please see the attachment.

Please see the attachment.
Author Response
Thank you very much for taking the time to review this manuscript. Please find the detailed responses below and the corresponding revisions in the re-submitted files.

Reviewer 2 Report
Tufarelli et al. evaluated the role and mechanism of retinol and its transformation product, retinoic acid, in modulating oxidative stress-induced damage to the duck intestinal epithelial barrier in vitro. The study is well-designed, and I recommend its publication after major revision.
1. Simple summary, line 16: Please specify which treatment
2. Abstract: not clear. I suggest covering in brief the following main items: aim of the study, methodology, main findings, and conclusion.
3. Line 27: Which genes? Please specify.
4. Line 28: Which barrier?
5. Line 30: which group
6. Line 34: oxidative should be Oxidative
7. Introduction: Lines 42-44: Please explain the impact of oxidative stress in more detail.
Please add the following paragraph: „Heat stress, dysbiosis, leaky gut syndrome, and mycotoxins are the main “secret killers” in poultry that lead to chronic oxidative stress and inflammation, which in turn impact health and animal performance. Additionally, chronic stress in poultry is linked with the emergence of antimicrobial resistance (AMR), which the WHO has recently identified to be among the most important problems threatening human health globally that increased the demand for safe antimicrobials to treat the collateral damages resulting from dysbiosis (https://doi.org/10.51585/gjvr.2022.3.0047). All forms of chronic stress, regardless of the origin, negatively impact the chicken's overall performance, health, and welfare (https://doi.org/10.51585/gjvr.2023.1.0051)
8. Line 84: Please briefly explain the process for preparing the cells.
9. Line 87: Please provide the supplier, City, and Company for all materials used in this study
10. Line 88: How long were cells incubated?
11. Line 94: Please expand the full name of TG1/RA
12. Line 97: GC2? Do you mean CG2?
13. Line 99: What is meant by cell status? Please specify.
14. Line 100: Cell counting for cell viability assessment is insufficient. Authors have to use specific tests to determine the cell viability, such as MTT. Also, cell proliferation needs to be assessed using a cell proliferation test.
15. Line 102: Again, what is meant by cell status?
16. Line 113-114: Please paraphrase
17. Line 114: ……..10 min, and take the supernatant should be …..10 min, afterwards, supernatants were collected.
18. Line 117: with …should be using
19. Line 134: Please delete: in a volume of 40 µL
20. Line 138: Premier should be primer
21. Figure 1. The photos are unfocused and require an increase in magnification
22. Table 2: Please expand VA/RA
23. Line 226: Please expand IEC
24. Figure 2 is missing??
25. Figure 7 is missing??
26. Figure 8: The legends are not readable
27. Line 531: Please write a paragraph describing the role of retinol as a promoter of antioxidants.
28. Conclusion: needs to be adjusted according to reflect the obtained results. The aim of the manuscript was to decrease oxidative stress and its consequences. Authors can start the conclusion in this direction....
29. Line 449: Vitamin should be vitamin
30. Line 656: Patents?? Please delete or explain if applicable or not.
Extensive editing of English language required
Author Response

(The authors gave the same response as above.)

Round 2
Reviewer 1 Report
Please see the attachment.

Author Response
Thank you very much for taking the time to review this manuscript. Please find the detailed responses below and the re-submitted files.

Reviewer 2 Report
The manuscript is significantly improved. I recommend "Accept in present form"
Author Response
Thank you very much for taking the time to review this manuscript.